**PLOS** NEGLECTED TROPICAL DISEASES

# Diagnosis of soil-transmitted helminth infections with digital mobile microscopy and artificial intelligence in a resource-limited setting

Johan Lundin [1,2]*, Antti Suutala[2], Oscar Holmström[2], Samuel Henriksson[1], Severi Valkamo[1], Harrison Kaingu[3], Felix Kinyua[3], Martin Muinde[3], Mikael Lundin[2], Vinod Diwan[1], Andreas Mårtensson[4], Nina Linder[2,4]

**1** Department of Global Public Health, Karolinska Institutet, Stockholm, Sweden, **2** Institute for Molecular Medicine Finland (FIMM), HiLIFE, University of Helsinki, Helsinki, Finland, **3** Kinondo Kwetu Hospital, Kinondo, Kwale County, Kenya, **4** Global Health & Migration Unit, Department of Women's and Children's Health, Uppsala University, Uppsala, Sweden

\* johan.lundin@ki.se

**Data Availability Statement:** All data required to evaluate the conclusions in the article are included in the manuscript and/or the supplementary

## Abstract

### Background

Infections caused by soil-transmitted helminths (STHs) are the most prevalent neglected tropical diseases and result in a major disease burden in low- and middle-income countries, especially in school-aged children. Improved diagnostic methods, especially for light intensity infections, are needed for efficient, control and elimination of STHs as a public health problem, as well as STH management. Image-based artificial intelligence (AI) has shown promise for STH detection in digitized stool samples. However, the diagnostic accuracy of AI-based analysis of entire microscope slides, so called whole-slide images (WSI), has previously not been evaluated on a sample-level in primary healthcare settings in STH endemic countries.

### Methodology/Principal findings

Stool samples (n = 1,335) were collected during 2020 from children attending primary schools in Kwale County, Kenya, prepared according to the Kato-Katz method at a local primary healthcare laboratory and digitized with a portable whole-slide microscopy scanner and uploaded via mobile networks to a cloud environment. The digital samples of adequate quality (n = 1,180) were split into a training (*n* = 388) and test set (*n* = 792) and a deep-learning system (DLS) developed for detection of STHs. The DLS findings were compared with expert manual microscopy and additional visual assessment of the digital samples in slides with discordant results between the methods. Manual microscopy detected 15 (1.9%) *Ascaris lumbricoides*, 172 (21.7%) *Tricuris trichiura* and 140 (17.7%) hookworm (*Ancylostoma duodenale* or *Necator americanus*) infections in the test set. Importantly, more than 90% of all STH positive cases represented light intensity infections. With manual microscopy as the reference standard, the sensitivity of the DLS as the index test for detection of *A.*

material. Additional data are available on request from the Data Access Committee (FIMM-DAC) at Institute for Molecular Medicine Finland (FIMM), University of Helsinki, Helsinki, Finland; fimm-dac@helsinki.fi. Further requests for sharing of deidentified data (digitized samples) will be considered by the FIMM-DAC abiding the following principles: data will be securely stored with appropriate documentation and not disposed into publicly accessible domains or otherwise shared without explicit permission from the FIMM-DAC, and data are only used with the aim to generate data for the public good.

**Funding:** This research was financially supported by The Erling-Persson Foundation (grant number 2021 0110) JL, Vetenskapsrådet (grant number 2021-04811) JL, Finska Läkaresällskapet r.f. JL, Medicinska Understödsföreningen Liv och Hälsa rf JL and Wilhelm och Else Stockmanns stiftelse JL. The funders had no role in study design, data collection and analysis, decision to publish, or preparation of the manuscript.

**Competing interests:** JL and ML are founders and co-owners of Aiforia technologies Plc. JL and AS reported having a patent for Mobile Microscope pending (no.WO2017037334A1; the invention is related to the use of fluorescence imaging filters combined with inexpensive plastic lenses; all rights are with the University of Helsinki) and JL having a patent for a slide holder for an optical microscope pending (no.WO2015185805A1; related to motorization of regular microscopes).

*lumbricoides*, *T. trichiura* and hookworm was 80%, 92% and 76%, respectively. The corresponding specificity was 98%, 90% and 95%. Notably, in 79 samples (10%) classified as negative by manual microscopy for a specific species, STH eggs were detected by the DLS and confirmed correct by visual inspection of the digital samples.

## Conclusions/Significance

Analysis of digitally scanned stool samples with the DLS provided high diagnostic accuracy for detection of STHs. Importantly, a substantial number of light intensity infections were missed by manual microscopy but detected by the DLS. Thus, analysis of WSIs with image-based AI may provide a future tool for improved detection of STHs in a primary healthcare setting, which in turn could facilitate monitoring and evaluation of control programs.

## Author summary

In this school survey conducted within a primary healthcare setting in rural Kenya, a deep-learning system (DLS) was developed for detection of eggs secreted in human stool from intestinal worms, so called soil-transmitted helminths (STHs). Infections caused by STHs are the most common neglected tropical diseases and a major cause of health impairment in low- and middle-income countries. Novel diagnostic methods are needed to improve disease control. In the current study, stool samples were collected from children attending local primary schools and digitized using a portable slide scanner and uploaded via mobile networks to a cloud repository. Using the digital samples, a DLS was trained and tested for the detection of eggs of the most common STHs (*Ascaris lumbricoides*, *Tricuris trichiura* and hookworms) and compared to conventional manual microscopy analysis of the samples. The results showed that the DLS was able to accurately detect STHs in the digital samples. Although the DLS detected some false positive parasites, it was able to detect a significant number of parasites that had been missed with conventional microscopy analysis. The study concludes that digital microscopy, supported by image-based artificial intelligence, can be implemented in a primary health care setting and may provide a future tool for improved STH detection and, thereby, better monitoring and evaluation of control programs. The digital method is especially promising in light intensity infections, with only a few parasite eggs per sample.

## Introduction

Neglected tropical diseases (NTDs) are a group of infectious diseases that affect more than a billion people globally and result in considerable negative health impacts in already resource-constrained areas [1]. The most prevalent NTDs are infections caused by the soil-transmitted helminths (STHs), which include the roundworms *Ascaris lumbricoides* (giant roundworm), *Trichuris trichiura* (whipworm), *Ancylostoma duodenale* and *Necator americanus* (hookworms). Together these pathogens cause chronic infections that result in disability, stigmatization, and significant economic consequences for societies [2,3]. As the STHs cause loss of micronutrients, they can cause neurocognitive problems, impaired growth and development and chronic fatigue in affected children [4]. Furthermore, the STHs are significant causes of morbidity and complications during pregnancy [5,6]. Current strategies to reduce morbidity

associated with STHs include mass drug administration (MDA) and interventions related to water, sanitation, and hygiene (WASH). The World Health Organization (WHO) has endorsed 2030 target goals for controlling STHs [7]. Currently, the recommended method of diagnostics of STHs is manual microscopy of stool samples to visualize and manually quantify parasites or parasite eggs [7,8]. However, there is a shortage of experts and access to microscopy equipment and laboratory infrastructure in regions with the highest STH prevalence [9,10]. Access to diagnostic tests is vital for efficient infection control and elimination of STHs as a public health problem, as well as and STH management, and there is a need for improved diagnostic tests that are accurate, feasible to use and deployable in regional laboratories, at the point-of-care (POC) and point-of-sampling in low-resource settings [11,12]. Although light microscopy has been widely implemented and validated, challenges with this method include the need for trained microscopists on-site and varying levels of sensitivity, especially in light-intensity infections [13]. The manual analysis of samples is furthermore time-consuming and labour-intensive, associated with inter- and intra-observer variability [14], and typically may require up to 10 minutes per slide [15].

There are a series of other diagnostic methods that have been evaluated, such as wet-mount methods, flotation-based methods (FLOTAC, Mini-FLOTAC, FECPAKG2, McMaster) and DNA-based methods (qPCR, LAMP) [16]. These have both advantages (typically higher sensitivity and specificity), but also significant disadvantages (lack of standardization, more time consuming, requirement for specialized equipment, higher costs) and manual Kato-Katz microscopy is still recommended by WHO for surveillance and epidemiologic surveys [1,7,8].

During the last decade, portable digital microscopes and cell phone-based microscopes have been increasingly explored as potential platforms for POC diagnostics of NTDs [17–21]. Studies have shown that the imaging performance achievable with these devices is sufficient to resolve the most common NTDs by both visual and automated analysis of the digitized samples [19,20]. Moreover, image-analysis methods, such as algorithms based on deep learning with artificial neural networks, have been applied to microscopy diagnostics for a large number of parasitic diseases [22] and shown to be suitable for diagnostics of NTDs with promising results [23–27]. Although the results demonstrate the potential for improved diagnostics of NTDs with these methods, clinical application of the methods in real-world settings has to date been limited and to our knowledge previously never been applied to assess diagnostic accuracy on a sample level based on digital scans of entire microscope slides, so called whole-slide images (WSIs).

In this study, we developed a digital diagnostic system for STHs and deployed it in a basic laboratory at a primary health care hospital in a rural region in Kenya with locally varying prevalence of STHs. We evaluated the system for detection of STHs in conventional Kato-Katz thick smears, prepared from stool samples of 1,335 children attending primary schools in the region. The samples were digitized with a portable WSI scanner, uploaded to a cloud-server over local data and mobile networks and used to train a deep learning system (DLS) to detect STHs. We evaluated the diagnostic accuracy of the DLS by comparison to expert manual microscopy assessment of the physical samples and to visual analysis of the digitized samples.

## Methods

### Ethics statement

Ethical approval for the current study (No: TUM ERC EXT/001/2020) was issued by the Chairperson of the Ethical Review Committee (Mombasa, Kenya) of the Technical University of Mombasa accredited by the National Commission for Science, Technology and Innovation Kenya. It was emphasized that participation in the study was voluntary and withdrawal

possible at any stage without further obligation. Informed verbal assent for study participation was acquired from participants and their parents or legal guardian, and a written consent form (in Swahili and English) was signed by the parents or legal guardian of the participant, as well as by the headmaster of the primary schools. Following the study, children in all participating schools were offered preventive treatment (albendazole at a dose of 400 mg), according to national guidelines. All participants with detected STHs were also examined by a clinician, besides prescribing the deworming medication. Treatment was offered by a designated clinician from the study hospital (Kinondo Kwetu Hospital). There was a follow-up on treated infected patients that was conducted by the Kwale County NTD Program coordinator who also provided the necessary drugs. The National NTD Program management was included in the treatment for effective follow up. Furthermore, the families of children with positive tests were also offered deworming treatment.

## Study design

This school survey was a diagnostic trial type I to establish the diagnostic accuracy of DLS with manual microscopy as the reference method [28], and reported in accordance with the STARD-guidelines (Fig 1) [29]. The study was based at a primary health care hospital (Kinondo Kwetu Hospital https://www.kinondokwetuhospital.com) in rural Kenya (Msambweni Subcounty, Kwale County). The hospital is fully owned and supported by a trust fund (Kinondo Kwetu Trust Fund) and serves the Kinondo region in Kwale County and provides basic health care, treatment of malaria, HIV, tuberculosis, parasitic diseases, antenatal care, deliveries, child care, vaccinations and health information. The laboratory has basic equipment including light microscopes, hematology, blood grouping, biochemistry and CRP analyzers, as well as rapid diagnostic test kits (e.g. dengue fever, HIV, hepatitis etc).

The target population was children aged between 5 and 16 years attending local schools in the region (Zigira, Masindeni and Kilole primary schools). The schools were invited to participate in the study, and eligible participants recruited from grades 1–8 after informed assent and consent had been obtained. Inclusion criteria were the following: age between 5 and 16, informed assent from the participant and consent from the parents/guardians acquired. Exclusion criteria were the following: Symptoms and signs of acute, severe disease, subject refuses consent, analysis of samples failed due to unsatisfactory quality of the physical or digital sample.

## Acquisition and preparation of samples

Stool samples were acquired from 1,335 children during a time of eight months (March to October 2020) with the assistance of a team of dedicated health care volunteers (HCVs). As the schools were temporarily closed due to the COVID-19 pandemic, study participants were provided with sample vials (30 mL) and instructions for sample collection and asked to collect the sample at home. Prior to the study, the research team based at the Kinondo Kwetu Hospital had briefed the team of HCVs with the intent of the study. Household mapping was done and each of the HCVs were allocated at least 20 households per day for the duration of the study. Sample vials were transported by the HCVs to the study laboratory at the Kinondo Kwetu Hospital for processing. During the study period, twenty samples were collected per day. Samples were processed in daily batches and Kato-Katz thick smears prepared by laboratory technologists following standard operating procedures (S1 Protocol). First, glass slides were labelled with a unique sample identifier number after which the sample was applied to the surface using a standardized wire mesh sieve for sample filtration. A standardized template was used to concentrate the sample in the centre of the slide. Following this, the sample was covered

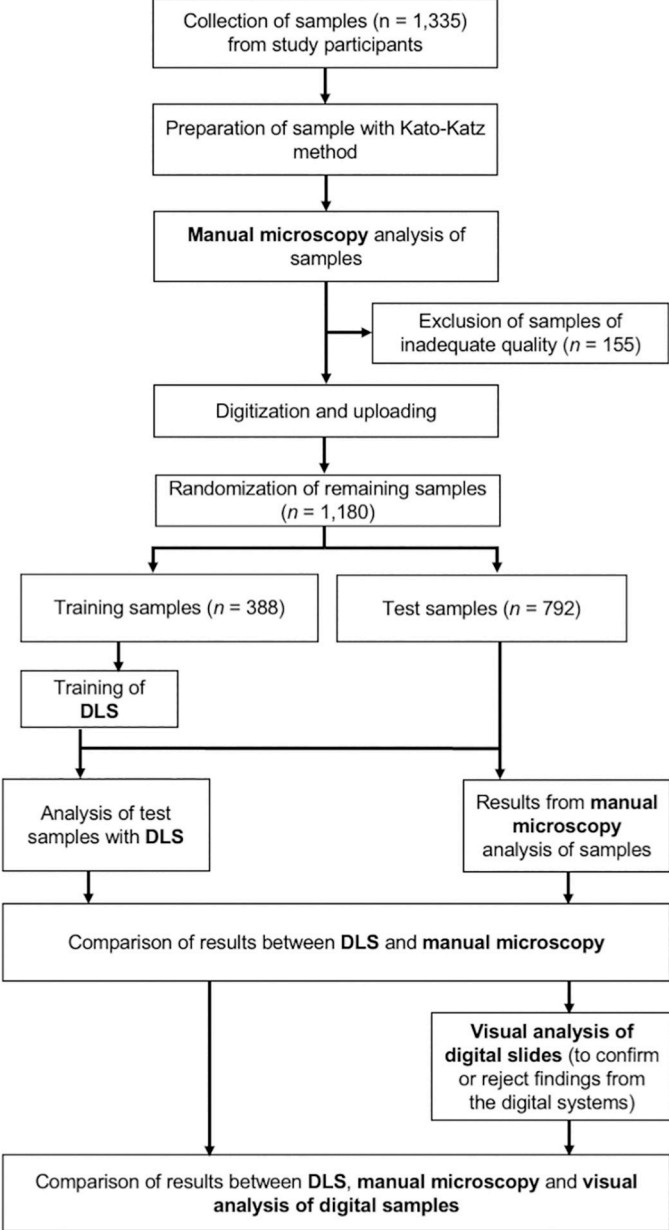

**Fig 1. STARD-style flowchart of study workflow.** STH = Soil-transmitted helminth. DLS = deep-learning system.

with a cellophane strip, immersed in glycerol-methylene blue solution (Kato's solution), placed on filter paper, and pressed to form an even surface between the cellophane and the slide. Slides were read immediately after preparation, to avoid disintegration of hookworm eggs [30], using a standard light microscope (CX23, Olympus, Tokyo, Japan). Slides were reviewed by a laboratory technologist with experience in diagnostics of STHs (FK), blinded from results of the digital analysis. Only one observer was used in order to allow sample analysis before

disintegration of hookworm eggs [8]. The entire cellophane-covered area was visually reviewed using 10x (NA 0.25) or 40x (NA 0.65) objectives and all visible parasite eggs of the species, *A. lumbricoides*, *T. trichiura* and hookworm eggs were recorded and counted. The Kato-Katz smear template is estimated to consist of 41.7 mg of stool. To calculate the number of eggs per 1 gram of stool (EPG) the egg count from the slide was multiplied by a factor of 24 (24 x 41.7 mg ≈ 1 g). Infection intensity was then categorized as light, moderate, and heavy infection according to the WHO guidelines [31]. Remaining stool of each sample was stored in a refrigerator for reference and discarded after a week on a 'first-in, first-out' basis.

Of the total number of 1,335 slides prepared, 155 (11.6%) were excluded due to inadequate quality for analysis and consequently 1,180 slides were included in the training and analysis of the method. Reasons for inadequate quality included insufficient amount of stool, sample contaminated with sand, too oily stool hence obscuring parasite eggs, stool too hard especially in constipation, fragmented filtrate which trapped water in the Kato-Katz smear causing areas of the sample to be empty or too dense, diarrhoea which could not form a firm smear, or high amounts of vegetable cells that obscured details including parasite eggs.

For development of the DLS, 388 (33%) slides were randomly selected to constitute the training set and the remaining 792 (67%) slides were used for testing of the algorithm. Manual microscopy assessment of samples was performed prior to the training and testing of the DLS, and results from the digital image analysis were, thus, not available during manual microscopy of the slides. Slides with discrepant results between the methods (manual microscopy and DLS-based analysis) in the test set were subjected to a visual review of the digital slides done by the expert microscopist (FK) and the researchers (AS, OH, SH, SV). During this phase visible STH eggs were detected in a number of samples that were not observed during the manual microscopy.

## Digitization of slides

Following preparation of slides and manual microscopy, the Kato-Katz thick smears were digitized with a compact and portable whole-slide scanner (Grundium Ocus 20, Grundium, Tampere, Finland) that weighs approximately 3.5 kilograms (Fig 2). The scanner digitizes samples with a 6-megapixel image sensor, using a 20× objective (NA 0.40) with a pixel size of 0.48 μm. The device is operated from a laptop computer over the wireless local area network. Coarse focusing of samples was performed manually on one field of view before initiation of the scanning which takes on average 5–10 seconds, whereafter the scanning process is fully automated using a motorized stage and a built-in autofocus routine. Scanning the entire standard sized Kato-Katz thick smear typically took 5–10 minutes. Once the scan is complete, the scanner automatically stitches all field-of-view images into a single WSI. Digital slides were transferred from the internal storage on the scanner to local storage on the laptop computer and to an external hard drive back-up unit in a Tagged Image File Format (TIFF). A desktop application (Aiforia Connector, Aiforia Technologies, Helsinki, Finland) was then used to convert the files to a wavelet file format (Enhanced Compressed Wavelet, Hexagon Geospatial, Madison, AL, USA) with a compression ratio (1:16) previously shown to preserve sufficient detail to not alter image-analysis results [32]. The same application automatically compresses the file into a single zip-file and uploads the file to an image-management and machine-learning platform (Aiforia, Aiforia Technologies, Helsinki, Finland). Uploading of slides was performed with the in-house ADSL connection and using a 3G/4G mobile-network router (Huawei B525S, Huawei Technologies, Shenzhen, China), operating on the local mobile network (Safaricom, Nairobi, Kenya). The average size of the digitized slides in the compressed format was 502 MB (range 127 MB–1,127 MB). Turnaround time for uploading of samples was 10–20 min per slide with the mobile network (upload speed 5–8 Mbps) or ADSL connection (upload speed

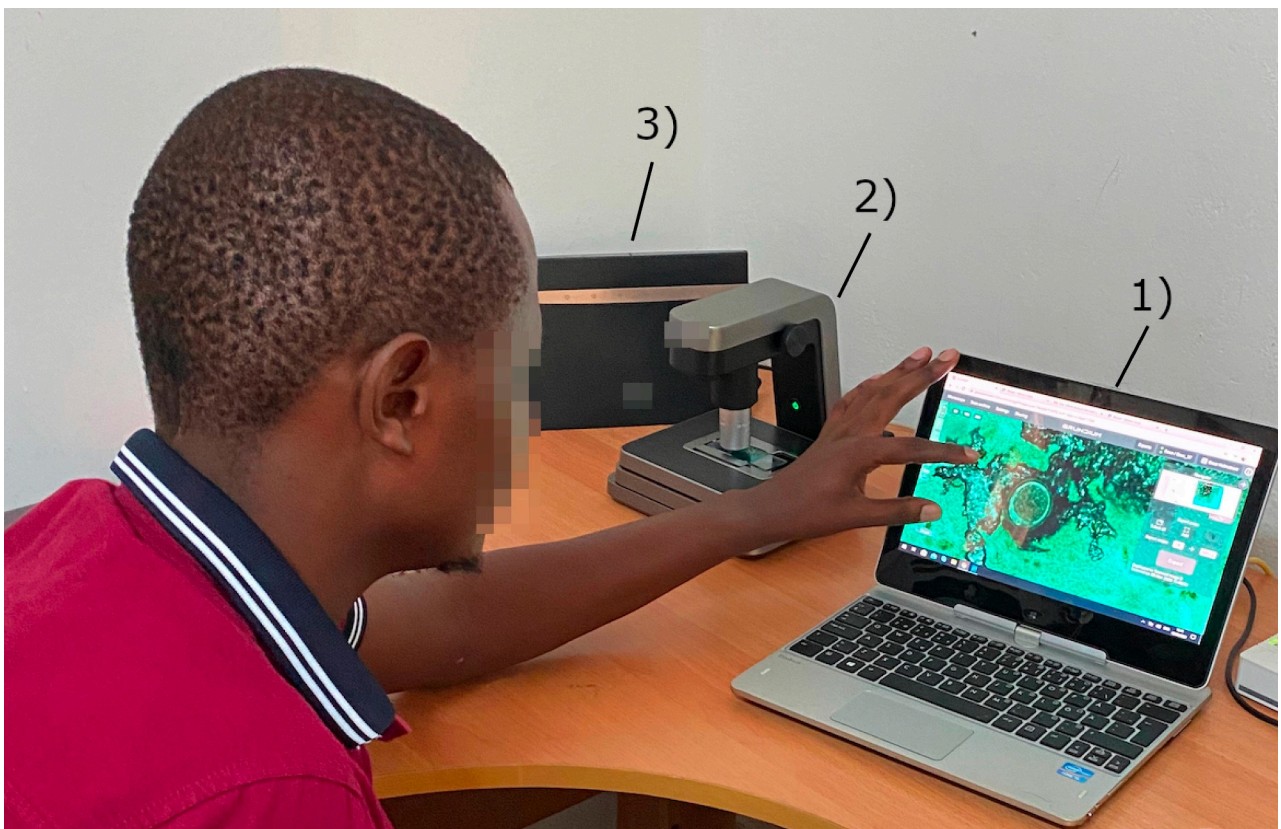

**Fig 2. Equipment and setup used for the digitization of microscopy slides in a primary health care setting.** Figure showing: **(1)** laptop computer used to operate the whole-slide scanner and manage the digital slides, **(2)** the whole-slide scanner used to scan slides and **(3)** a 3G/4G mobile network router used to upload scanned slides.

5–10 Mbps). On the cloud server, the slides were stored as JPEG-compressed tiles sized 512x512 pixels with pyramid-structure zoom levels (70% JPEG-quality). A typical WSI comprises approximately 10,000 tiles on the server. Access to the image server for slide viewing was established with a web browser using secure socket layer (SSL) encryption.

### Training of the deep learning system

For the development of the DLS used for the analysis of the digital samples to detect STHs, we utilized supervised deep learning. The image analysis was performed by an algorithm utilizing two sequential convolutional neural networks to detect STH eggs and classify them according to the different species (*A. lumbricoides*, *T. trichiura* and hookworm). Here, the first network (the "detector network") initially detected all potential STH egg candidates and forwarded the results to a second network (the "classifier network"), which then classified the findings into one of four categories: *A. lumbricoides*, *T. trichiura*, hookworm or artefact (i.e., debris or other non-STH objects). This dual-step operation principle of the DLS is illustrated in Fig 3.

Digital samples were acquired by downloading the scanned whole-slide data from the image management platform (Aiforia Hub, Aiforia Technologies Oy, Helsinki, Finland), and each slide was converted into a single TIFF file for further processing and algorithm training in MATLAB (MathWorks Inc, Natick, MA). Digital samples in the training series (n = 388) were visually reviewed and objects of interest (parasite eggs) in the samples from the different categories manually annotated to constitute the ground truth for the training data. An initial

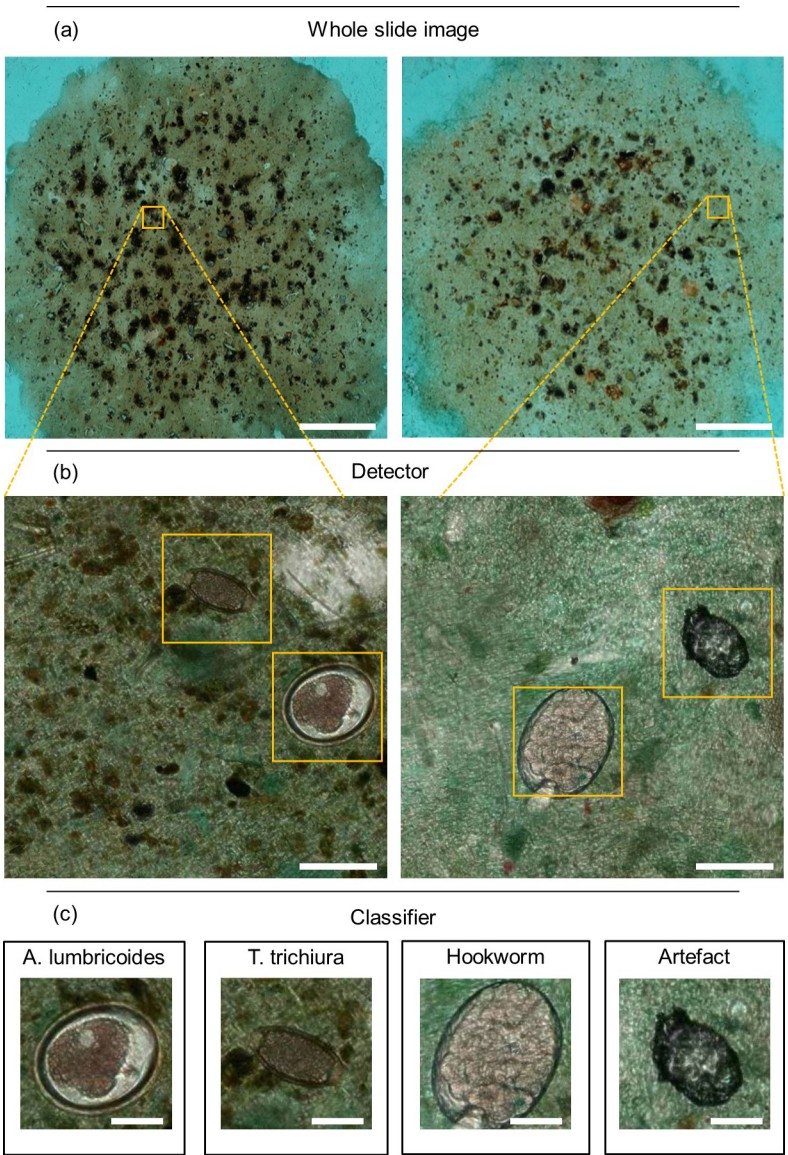

**Fig 3. Analysis of digitized whole slide images with the deep learning system.** Panels showing (a) the digitized samples, (b) first step: localization of parasite egg candidates and (c) second step: classification of the parasite egg candidates into the soil-transmitted helminth and artefact categories. Scale bars: (a) 5 mm, (b) 50 μm, (c) 25 μm.

set of 1,339 annotations were made by browsing the whole slide images of the training set and manually annotating visible *A. lumbricoides* (n = 371), *T. trichiura* (n = 530), hookworm (n = 51), and artefact (n = 446) findings. The initial annotations were done by the researchers (AS, OH, SH, SV), after thorough training on the morphology of the STHs provided by the laboratory technologist (FK) who performed the manual microscopy. After that, the rest of the object-level annotations were done by utilizing machine learning-assisted annotation (Fig 4). In this process, the image data in the training set was analysed by a previously trained algorithm, which then suggested undetected new potential objects of interest (e.g., parasites or artefacts) for the user to confirm or reject. The machine-learning annotations were completed by one researcher (AS) and in cases of uncertainty, confirmation was reached after consultations with the other researchers. These confirmed annotations (objects of interest and their

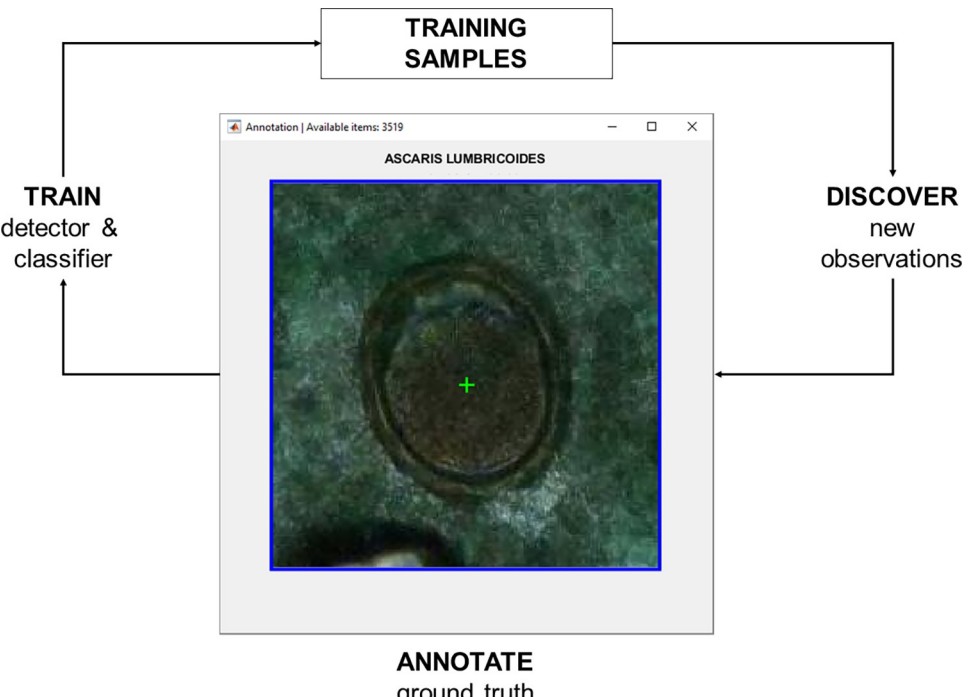

**Fig 4. Machine learning assisted annotation.** Figure depicts how the algorithm automatically suggests new, undetected objects of interest with associated labels for the user to confirm or reject. These findings are then included in the training data, but also used to improve the performance of the algorithm itself to suggest new objects more accurately for the user.

locations in the image) were then incorporated into the training data for the DLS, improving the algorithm's performance. This iterative annotation process was carried on until no new acceptable findings were found in the training slides. In total, a final set of 15,058 annotations, with manual and AI-assisted annotations for *A. lumbricoides* (n = 2,299 from 39 slides), *T. trichiura* (n = 2,727 from 94 slides), hookworm (n = 552 from 38 slides) and artefacts (n = 9,480 from 384 slides) were available for training the DLS.

For the object detector in the DLS, the you-only-look-once version 2 (YOLOv2) algorithm was used [33]. This algorithm predicts class probabilities for objects within bounding boxes in images. For feature extraction in the network, we utilized a pre-trained ResNet50 network [34]. Training of the algorithm was performed using training regions with a size of 512 x 512 $px^2$ (246 x 246 μm) containing the objects of interest annotated in the training data. Training was performed with 20 training epochs with a minibatch size of 25, using a stochastic gradient descent solver with a momentum of 0.9 and initial learning rate of 0.0005.

For the object classification, transfer learning [35] was utilized using a pre-trained ResNet50 network. The algorithm was a multiclass classification convolutional neural network which was trained using manually selected image regions measuring 150 x 150 $px^2$ (72 x 72 μm) in the training samples. These areas represented visible *A. lumbricoides*, *T. trichiura*, hookworm and artefacts. Training of the classifier was performed in 50 training epochs with a minibatch size of 32, using a stochastic gradient descent solver with a momentum of 0.9 and initial learning rate of 0.003 and dropping the learning rate by half after every 10 epochs. Image augmentations of the training data were utilized to increase the generalizability of the model and to prevent overfitting for both networks in the DLS. For the detection network, augmentations used were variation of scale (±10%), XY reflections, saturation offset (0.1), brightness offset

(0.1), hue offset (0.05), and scale factor of contrast (0.1). The object classifier training data was augmented using variation of scale (±5%), rotation (0–360˚), XY shear (±5˚), XY reflections, XY translations (±10 pixels), saturation offset (0.1), brightness offset (0.15), hue offset (0.05), and scale factor of contrast (0.15).

Analysis of samples in the test set was performed using a 512 x 512 px$^2$ (246 μm x 246 μm) sized scanning window with an overlap of 128 pixels (61 μm) with the adjacent regions to cover the whole sample image area. Overlapping bounding boxes of the detected objects were eliminated by selecting the box with the highest confidence (i.e. probability score). The detected bounding box was then resized to 150 x 150 px$^2$ (72 μm x 72 μm), and that subregion image was forwarded to the second network for classification. The threshold for the confidence was set to 50/100 to reduce the initial false positive rate and computation cost. Similarly, a threshold for the parasite egg finding in the classifier was set to 90/100. We achieved an analysis speed of approximately 14 digital samples (whole-slide images measuring 25 x 25 mm$^2$) per hour using a computer equipped with an Intel Xeon E3-1241 v3 processor, a NVIDIA GeForce GTX 1660 Super graphics processing unit, and 32GB of RAM running MathWorks MATLAB R2020b on the Microsoft Windows 10 operating system. If one WSI in average consists of 10,000 tiles sized 512x512 pixels, this corresponds to approximately 140,000 tiles (fields-of-view; FOV) per hour and 2,333 tiles (FOVs) per minute.

## Statistical analysis

Results from analysis of samples were entered into a standardized spreadsheet table (Microsoft Excel, Microsoft, Redmond, WA, USA). We used a general-purpose statistical software suite (Stata 15.1, Stata, College Station, TX, USA) for analysis of the results. The sample sizes used in the study were determined based on calculations with a previously described sample size formula [36], assuming an estimated overall infection prevalence (prevalence of STHs in the population, $P_r$) of 20%, an $\alpha$ level of 0.05 and a precision parameter ($\epsilon$) of 0.05 to determine whether sensitivity ($S_N$) and specificity ($S_P$) were equivalent to the reference standard (expert manual microscopy assessment of samples).

$$n_{Sensitivity} = \frac{Z_{1-\alpha/2}^2 S_N (1 - S_N)}{\varepsilon^2 \times P_r}$$

$$n_{Specificity} = \frac{Z_{1-\alpha/2}^2 S_p (1 - S_p)}{\varepsilon^2 \times (1 - P_r)}$$

Based on these calculations, sample sizes of 692 (for sensitivity) and 173 (for specificity) were required, assuming a predetermined level of sensitivity and specificity of 0.90 [37]. All performed statistical tests were two-sided unless otherwise stated. Diagnostic accuracy was evaluated in terms of sensitivity, specificity, positive and negative predictive values. To measure the correlation between the manual and DLS-based counting of EPG, we calculated the Pearson correlation coefficient. The level of statistical significance was 0.05 and statistical estimates of diagnostic accuracy were reported with 95% confidence intervals (CI95%).

## Results

### Prevalence of soil-transmitted helminths

After exclusion of slides used for training (n = 388) of the DLS and slides with unsatisfactory quality, 792 slides remained for analysis (Fig 1). Prevalence of the STHs according to manual microscopy and the DLS are presented in Table 1.

**Table 1. Prevalence of soil-transmitted helminths.**

|  | Manual microscopy | Deep learning system |
|---|---|---|
| Soil -transmitted helminths (any species) | 275 (34.7%) | 318 (40.2%) |
| *A. lumbricoides* | 15 (1.9%) | 26 (3.3%) |
| *T. trichiura* | 172 (21.7%) | 222 (28.0%) |
| Hookworm | 140 (17.7%) | 137 (17.3%) |
| Mixed infections | 49 (6.2%) | 62 (7.8%) |
| Negative | 517 (65.3%) | 474 (59.8%) |

Prevalence of soil-transmitted helminths in the test set of 792 slides, as detected by expert manual microscopy and by analysis of the digital slides by the deep-learning system.

## Diagnostic accuracy for detection of soil-transmitted helminths by the deep learning system

The sensitivity, specificity, positive and negative predictive values for the detection of STHs by the DLS, as compared to manual microscopy assessment of samples are presented in Table 2.

Discrepancy between the methods was observed especially in the analysis of hookworm samples, where 33 of samples were false negative by the DLS, in which hookworm eggs had been detected by manual microscopy. The largest number of false positive samples was related to detection of *T. trichiura*, where the DLS detected parasites in 64 samples classified as negative by manual microscopy (Table 3).

## Visual review of digitized samples

A review of digitized samples in case of discordant results between manual microscopy and analysis of digital samples by the DLS was performed by the researchers (AS, SV, SH, JL) independently of the laboratory technologist who performed the manual microscopy (FK). A total of 5 of 14 (36%) samples positive for *A. lumbricoides*, 59 of 64 (92%) samples positive for *T. trichiura*, and 15 of 30 (50%) positive for hookworm according to the DLS but negative according to manual microscopy, were confirmed positive in the digital slides. A re-assessment of the diagnostic accuracy was performed after visual review of the digital slides, with a combined manual microscopy and visual review as the reference standard and is reported in Supporting Table 1. Examples of the visually verified STHs are shown in Fig 5. Thus, in a total of 79 (10%) out of 792 samples classified as negative for a specific STH species by manual microscopy assessment, the DLS diagnosis could be confirmed as correct. With visual review of digital slides as a corrected reference standard, the most prominent change in diagnostic accuracy was the increase in specificity for *T. trichiura* from 89.7% of 99.1% (S1 Table). The remaining samples classified as positive by the DLS, but negative according to manual microscopy often displayed artefacts with similar morphologic features as the parasite eggs (S1 Fig).

**Table 2. Diagnostic accuracy of the deep learning system.**

|  | Sensitivity, % (CI95%) | Specificity, % (CI95%) | Positive predictive value, % (CI95%) | Negative predictive value, % (CI95%) |
|---|---|---|---|---|
| *A. lumbricoides* | 80.0 (51.9–95.7) | 98.2 (97.0–99.0) | 46.2 (26.6–66.6) | 99.6 (98.9–99.9) |
| *T. trichiura* | 91.9 (86.7–95.5) | 89.7 (87.0–92.0) | 71.2 (64.7–77.0) | 97.5 (95.1–95.9) |
| Hookworm | 76.4 (68.5–83.2) | 95.4 (93.5–96.9) | 78.1 (70.2–84.7) | 95.0 (93.0–96.5) |

Results from the analysis of samples by the deep learning system, as compared to the manual microscopy assessment of slides for the detection of soil-transmitted helminths. CI = confidence interval.

**Table 3. Agreement between the deep learning system and manual microscopy.**

| | | Analysis by the deep learning system (DLS) | | |
| --- | --- | --- | --- | --- |
| | | Negative | *A. lumbricoides* | Total |
| Manual microscopy | Negative | 763 | 14 | 777 |
| | *A. lumbricoides* | 3 | 12 | 15 |
| Total | | 766 | 26 | 792 |
| | | Analysis by DLS | | |
| | | Negative | *T. trichiura* | Total |
| Manual microscopy | Negative | 556 | 64 | 620 |
| | *T. trichiura* | 14 | 158 | 172 |
| Total | | 570 | 222 | 792 |
| | | Analysis by DLS | | |
| | | Negative | Hookworm | Total |
| Manual microscopy | Negative | 622 | 30 | 652 |
| | Hookworm | 33 | 107 | 140 |
| Total | | 655 | 137 | 792 |

Contingency table showing the results achieved by analysis of samples with manual microscopy and by analysis with the deep learning system. Hookworm = *Ancylostoma duodenale* or *Necator americanus*.

All the DLS false negative samples i.e. three *A. lumbricoides*, 14 *T. trichiura*, and 33 hookworm samples that were recorded as positive according to manual microscopy represented light intensity infections as reported in detail below. In the visual assessment of the DLS false negative samples, glycerol disintegration of hookworm eggs had affected 15 of 33 (45%) samples, as partly disintegrated hookworm eggs which had not been detected by the DLS could be observed in the digital images (S2 Fig). DLS false negative samples typically also had out-of-focus areas or dense parts that could explain that a few eggs seen in manual microscopy were missed by the DLS.

## Object-level detection of parasites in the samples by analysis with the deep learning system

Following the sample-level analysis of results, we also compared the number of individual parasite eggs in the samples where results were available for both manual and DLS-based quantification of parasites (*n* = 241). Egg counts according to species and detection method are shown in Table 4. Overall, 93% of *A. lumbricoides*, 93% of *T. trichiura* and 97% of hookworm samples in the test set represented light intensity infections according to manual microscopy.

A majority (*n* = 170, 72%) of the samples showed higher STH egg counts when analysed by the DLS, compared to the STH egg counts recorded by manual microscopy. Especially for *T. trichiura*, the EPGs according to the DLS were higher than for manual microscopy for both light and moderate intensity infections (S3 Fig). The correlation between the number of detected STH eggs by manual microscopy and DLS was significant (r = 0.63, p = 0.04). When infection intensity was categorized as light, moderate, and heavy infection according to the WHO guidelines, one case of *A. lumbricoides* (10%), 13 of *T. trichiura* (10%) and one case of hookworm infection (1%) changed from light to moderate when analysed with the DLS as compared to manual microscopy, whereas three cases of hookworm infection changed from moderate to light (3%) according to the DLS (Table 5).

For the samples classified as false negative by the DLS, the average number of STH eggs according to manual microscopy was 1 for *A. lumbricoides* (24 EPG), 5 (110 EPG) for

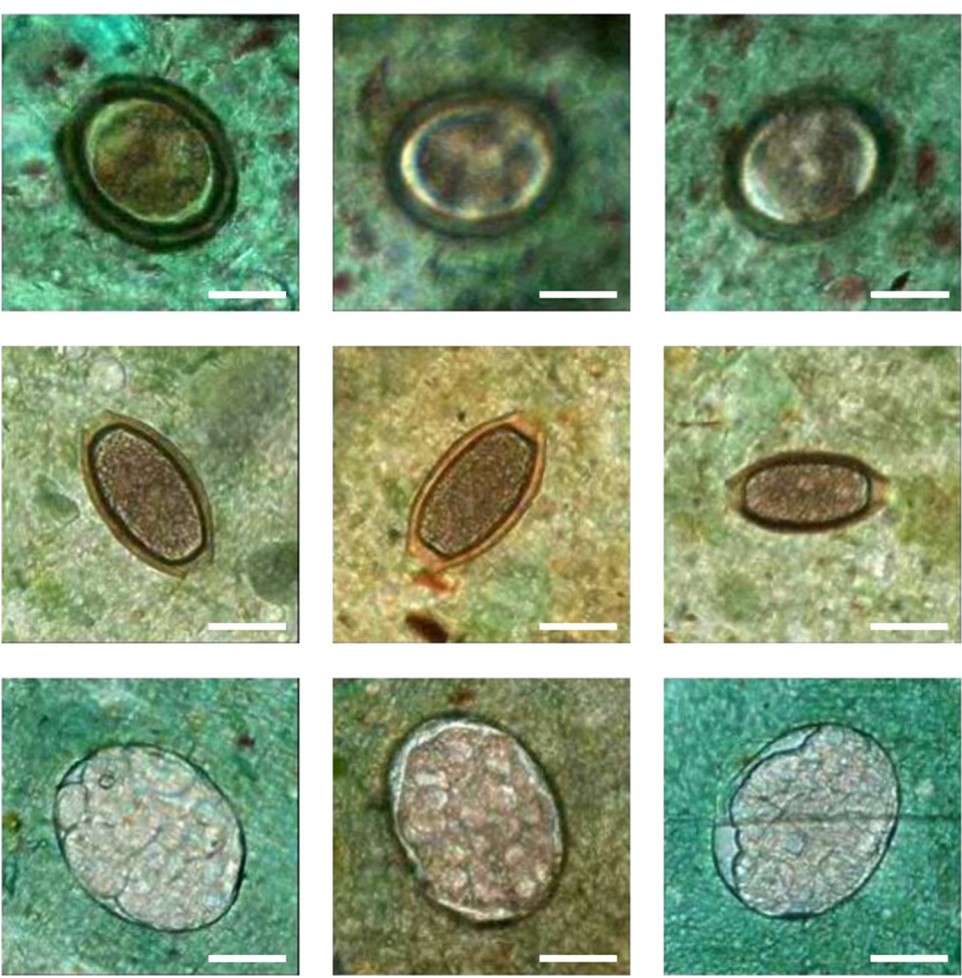

**Fig 5. Examples showing visually verified soil-transmitted helminths.** Visually verified soil-transmitted helminth eggs detected by the deep learning system from slides that were originally classified as negative by manual microscopy: *A. lumbricoides* (top row), *T. trichiura* (middle row), and hookworm (bottom row) findings. Scalebar 25 μm.

*T. trichiura*, and 5 for hookworm (129 EPG), all representing light intensity infections. When assessing the object-level findings for the samples classified as false positive by the DLS, we also observed a low level of detected STH eggs in the samples (mean = 4, range 1–16), corresponding to 96 EPG (24–384).

**Table 4. Quantification of parasite eggs.**

| Species | Method | Average number of parasite eggs per sample | Range | Standard deviation | Average eggs per gram |
|---|---|---|---|---|---|
| *A. lumbricoides* | Manual microscopy | 69 | (1–364) | 119.7 | 1,656 |
|  | DLS | 248 | (1–1895) | 598.1 | 5,952 |
| *T. trichiura* | Manual microscopy | 11 | (1–130) | 19.5 | 264 |
|  | DLS | 27 | (1–288) | 46.9 | 648 |
| Hookworm | Manual microscopy | 15 | (1–122) | 22.6 | 360 |
|  | DLS | 13 | (1–108) | 18.3 | 312 |

Quantification of parasite eggs in slides classified as positive by both manual microscopy and by the deep learning system (DLS).

**Table 5. Classification of the intensity of soil-transmitted helminth infections.**

| | | | Deep learning system | | |
|---|---|---|---|---|---|
| | | | **Light** | **Moderate** | **Heavy** |
| *A. lumbricoides* | Manual microscopy | Light | 8 | 1 | 0 |
| | | Moderate | 0 | 1 | 0 |
| | | Heavy | 0 | 0 | 0 |
| | | | Light | Moderate | Heavy |
| *T. trichiura* | Manual microscopy | Light | 133 | 13 | 0 |
| | | Moderate | 1 | 10 | 0 |
| | | Heavy | 0 | 0 | 0 |
| | | | Light | Moderate | Heavy |
| Hookworm | Manual microscopy | Light | 102 | 1 | 0 |
| | | Moderate | 3 | 1 | 0 |
| | | Heavy | 0 | 0 | 0 |

Intensity of infections in the test set where results were available for both manual and deep learning system-based quantification of parasite eggs.

## Discussion

Analysis of digitally scanned WSIs of Kato-Katz thick smears with the DLS provided high diagnostic accuracy of the most common STHs. Importantly, a substantial number of light intensity infections were missed by manual microscopy but detected by DLS supported digital microscopy, which challenges the role of expert manual microscopy as the recommended method for detection of STHs. Also, STH egg counts were in average significantly higher assessed by the DLS as compared to manual microscopy. Samples were digitized with a portable digital whole slide scanner and uploaded via local data and mobile networks to a cloud server for analysis with the DLS. The DLS was trained with a subset of the samples to detect the most common STHs (*T. trichiura*, *A. lumbricoides* and hookworm) in the digital samples. With manual microscopy as the reference standard, a high level of sensitivity (76–92%) and of specificity (90–98%) was observed for the DLS for the detection of STHs. After visual review of samples with discrepant results between the manual and digital methods, the DLS yielded a low proportion of false positive (1–2%) and false negative (0.4–4%) cases. The discrepant results typically originated from samples with low numbers of visible STH eggs and containing artefacts showing similar morphological features as the parasites. Notably, the DLS detected a high number of samples (n = 79) with STHs which had not been detected by manual microscopy. These samples also primarily represented slides with low numbers of visible parasite eggs, which have previously been recognized as challenging to analyse by manual microscopy [12,13].

Overall, for the DLS, the detection of hookworm eggs yielded the highest discrepancy when compared to manual microscopy and the lowest sensitivity, although specificity was high. As the slides had been digitized with a delay following manual microscopy, a possible contributing factor could be the disintegration of hookworm eggs [37]. This is also supported by the fact that partly disintegrated hookworm eggs could be visually identified in several digital slides classified as positive for hookworm by manual microscopy. By ensuring more rapid sample processing and including also non-intact hookworm eggs in the training data, the accuracy for detection of hookworm could likely be improved. Another potentially contributing factor may be the lower number of manual annotations of hookworm eggs available in the training data (n = 552), compared to the other parasites (n > 2000). On an object-level, we observed a clearly higher number (approximately three times higher) of detected *A. lumbricoides* and *T.*

*trichiura* parasite eggs by DLS assessment of slides compared to manual microscopy. Even though the DLSs detected generally higher number of parasite eggs in the positive samples, a significant correlation between the manual and digital results were observed. The manual quantification of individual parasites in large sample areas is cumbersome, and our results suggest that the DLS-based analysis could improve the counting process. An additional explanation for higher counts and detection of initially missed eggs of *A. lumbricoides* and *T. trichiura* in the digital samples could be a longer clearing time. However, while hookworm eggs may disintegrate over time, *A. lumbricoides* and *T. trichiura* egg counts have also been shown to remain relatively stable up to 50 hours after preparation [38]. The potential effect of time from preparation to analysis on diagnostic accuracy of the digital method should be assessed in future studies, for example by randomizing the Kato-Katz thick smears to either manual microscopy or DLS analysis or to prepare duplicate slides which are processed in parallel with the two methods.

The results from our study support findings from earlier work, where deep learning-based algorithms combined with digital microscopy has shown potential for diagnostics of NTDs and STHs [20,23,25]. A major strength of the current study is that true whole-slide images of samples scanned in a primary healthcare setting in an STH endemic setting were analysed, as compared to several previous studies that have only analysed cropped, smaller areas of samples [23,24,26,27]. We found only one previous study where the entire sample area was digitized in a subset of the samples, but the diagnostic accuracy on a sample level was not reported in that study [25]. Although these results have been encouraging, assessment of diagnostic accuracy requires the analysis of whole slide images (i.e., the entire slide) and a sample-level diagnosis. Here, we obtained a large number (n = 1,335) of conventionally prepared Kato-Katz thick smears and digitized and assessed the diagnostic accuracy of the AI-based method based on the entire slides. As the samples were collected, processed, digitized, and uploaded at a basic laboratory in a primary health care setting, the findings demonstrate how the method can be implemented also in a rural, real-world setting, where access to advanced infrastructure for digital microscopy (such as conventional whole slide scanners) is not available. We were not able to find any previous studies that have applied AI-based methods in a similar basic laboratory facility in a primary health care setting. Moreover, the number of slides analysed in this study exceeds previous similar work with a large margin.

WHO has published a target product profile (TPP) for monitoring and evaluation of STH control programmes [11]. The target product is an in vitro/ex vivo laboratory-based (minimum) or point-of-sampling (ideal) test that allows for quantitative detection of analytes specific to STHs in all age groups. We compared our method to the TPP and found that it fulfils most of the WHO stipulated criteria for a laboratory-based test: Our method can be used in regional or national diagnostic testing laboratories by trained laboratory technicians ($<$ 1-week training); is portable (weighs less than 5 kg including a laptop computer and router), is specific ($\geq$ 94%) to each *A. lumbricoides*, *T. trichiura* and hookworm and has a sensitivity of at least 60% for each of the three species, if the combined manual microscopy and visual analysis of digital slides is used as the reference standard. However, the diagnostic accuracy of our method as compared to more sensitive and specific STH diagnostic methods (i.e. closer to a 'ground truth') such as PCR-based methods or similar remains to be established. If run on battery, the method also has the potential to be used as a point-of-sampling test, where health personnel and community health workers should be able to perform and interpret the test with only a single day of training [11]. Further, the TPP test for point-of-sampling should allow for a throughput of at least seven samples per hour and its cost should not exceed 3 USD per test. Further studies are needed to assess if our method could meet these requirements.

There are limitations of our study that need to be considered when interpreting the results. Firstly, the microscopy assessment of samples here was performed by only a single expert due to practical reasons (such as challenges related to preservation of Kato-Katz thick smears). The visual evaluation of microscopy samples is known to be subjective [39] and, therefore, prone to variations in sensitivity, especially in light intensity infections [13]. Multiple readers per slide would therefore likely improve the reliability of the reference standard. Also, other diagnostic methods, such as DNA-based methods could be used as the reference standard [16]. As the samples here were prepared at a single laboratory, external validation of the results is also needed to assess generalizability to other settings. Furthermore, as the digitization of samples was performed with a delay after the manual analysis of samples, we consider it likely that the disintegration of hookworm eggs or clearing of *A. lumbricoides* and *T. trichiura* eggs could have affected the results. Finally, although the total number of samples showing STH infection was relatively high, the distribution of the different STH species varied due to local prevalence, and therefore we obtained variable numbers of samples representing the different species (e.g. the number of samples positive for *A. lumbricoides* was lower compared to the other species), which could affect the results. Also of note is that 93–97% of all analysed samples in the test set represented light intensity infections, with very few detected STH eggs per sample.

As the performance of this diagnostic method is also affected by several other factors than the analysis of samples, such as sample collection, preparation, digitization, and data upload future studies should also focus on these steps. Here, for example, the analysis of thicker sample areas proved to be challenging both in manual microscopy and digital analysis of samples, and thus emphasizes the importance of high-quality sample preparation. As a subset of digital samples demonstrated certain areas out of focus and with uneven brightness, scanning of multiple focal planes (e.g., z-stacks) and imaging with high dynamic range could likely also contribute to improved diagnostic accuracy.

Also, the current method relies on internet connectivity, since images are transferred to a cloud environment. Sufficient internet connections might not be available in all rural areas. However, in our studies in rural Kenya and Tanzania [27,40], connectivity has been sufficient and upload of data has not been a bottleneck with speeds of 5–10 megabits per second. There are several advantages of data upload to a central repository in a cloud environment, including opportunities to have human experts in the loop for remote verification, to create large-scale databases, continuously monitor the accuracy of the algorithms and quality of samples, perform surveillance of the target infections and to collect more training data. Ideally, the algorithms should also be applicable without internet connectivity and to run on local devices, but also in that case sharing and transferring data when mobile networks are available should be considered an advantage. In a study on deep learning based detection of *Schistosoma haematobium* in urine samples, an algorithm was developed that was run locally and reached a high diagnostic accuracy on an image (e.g. tile or field-of-view) level [41]. The computational speed was approximately 10 images per minute (sized 1,520 x 2,028), equivalent to 0.5 megapixels per second as compared to approximately 14 megapixels per second with our method (2500% faster). This means that analysis of one of our WSIs with the described local solution would take almost 2 hours, as compared to 15–25 minutes (upload 10–20 minutes + analysis 5 minutes) with our solution.

The total turnaround time of 20–35 minutes per sample (scanning 5–10 minutes, upload 10–20 minutes and analysis 5 minutes) with our method could be considered a limitation, but there are several opportunities to make the current solution faster. These include more efficient hardware and software, either running locally or in a cloud environment. Also, the speed of mobile networks is likely to improve and access to 5G networks could allow 10-fold faster upload to a cloud environment, which would translate into a 1.5–2.5 minute upload time [42].

In the current study, we decided to use the magnification of 20x that is standard among many commercial microscope scanners and that can serve additional diagnostic purposes such as screening for cervical pre-cancerous lesions in pap smears [40]. It might be feasible to identify parasite eggs using a lower magnification, and a 10x magnification could result in a WSI that is one-quarter the size, thereby further speeding up the processing time. However, the diagnostic accuracy of STH detection using a lower magnification and resolution needs to be assessed in future studies.

Overall, the findings of this study support results of previous studies on AI-based detection of STHs, and adds evidence of feasibility of the use of portable whole-slide digital microscopy scanners combined with deep learning-based analysis of samples for the diagnosis of STHs. Although here we only investigated the most common STH species, this technology can be applied to microscopy diagnostics of other similarly diagnosed infectious diseases, such as other NTDs. Further studies should focus on multi-centre, prospective evaluation of the generalisability of the method, and evaluation of other potential diagnostic targets.

## Conclusion

Deep learning-based detection of common soil-transmitted helminths in WSIs of Kato-Katz thick smears is feasible and can reach high diagnostic accuracy using a minimal digital microscopy infrastructure approach that can be applied in basic laboratory facilities in primary health care setting of STH endemic countries. Kato-Katz thick smears can be digitized with a portable slide scanner and uploaded to a cloud environment via a basic laptop computer and local data networks for AI-based analysis. This method could be used to facilitate the diagnostics of STHs in light intensity infections and is also likely to be applicable in field settings, and to digital diagnostics of other conditions, such as other NTDs.

## Supporting information

**S1 Fig. Examples of false positive objects.** Examples of artefacts classified as parasite eggs by the deep learning system causing a false positive result: (a) *A. lumbricoides*, (b) *T. trichiura*, and (c) hookworm (c) findings.
(TIF)

**S2 Fig. Examples of false negative objects.** Examples of desiccated hookworm eggs found in the digital images causing false negative results in the deep learning system (DLS) analysis.
(TIF)

**S3 Fig. Scatter plots of eggs per gram (EPG) as quantified by manual microscopy and the deep learning system (DLS) in slides classified as positive by both manual microscopy and the DLS.** The x-axis represents manual microscopy results, while the y-axis depicts the DLS results, separately for a) *A. lumbricoides*, b) *T. trichiura*, and c) hookworm. Infection intensity levels for each species are denoted by letters on the plots (L for light, M for moderate, and H for heavy infection).
(TIF)

**S1 Protocol. Standard operation procedure for preparation of Kato-Katz thick smears.**
(DOCX)

**S1 Table. Diagnostic accuracy for soil-transmitted helminths (STHs) after visual review of the digital samples, i.e. with manual microscopy and visual analysis of digital slides as the reference standard.** CI = confidence interval.
(DOCX)

## Acknowledgments

We thank the Institute for Molecular Medicine Finland (FIMM) Digital Microscopy and Molecular Pathology Unit (University of Helsinki) and Biocenter Finland (Helsinki Institute of Life Science HiLIFE, University of Helsinki) for access to image management and processing infrastructure during the study. We thank Hakan Küçükel (FIMM) for IT support.

## Author Contributions

**Conceptualization:** Johan Lundin, Antti Suutala, Oscar Holmström, Samuel Henriksson, Severi Valkamo, Harrison Kaingu, Mikael Lundin, Vinod Diwan, Andreas Mårtensson, Nina Linder.

**Data curation:** Johan Lundin, Antti Suutala, Oscar Holmström, Samuel Henriksson, Severi Valkamo, Felix Kinyua, Martin Muinde, Mikael Lundin, Nina Linder.

**Formal analysis:** Johan Lundin, Antti Suutala, Oscar Holmström, Samuel Henriksson, Severi Valkamo, Felix Kinyua, Nina Linder.

**Funding acquisition:** Johan Lundin, Harrison Kaingu, Nina Linder.

**Investigation:** Johan Lundin, Antti Suutala, Oscar Holmström, Samuel Henriksson, Severi Valkamo, Harrison Kaingu, Felix Kinyua, Martin Muinde, Mikael Lundin, Vinod Diwan, Andreas Mårtensson, Nina Linder.

**Methodology:** Johan Lundin, Antti Suutala, Oscar Holmström, Samuel Henriksson, Severi Valkamo, Harrison Kaingu, Felix Kinyua, Mikael Lundin, Nina Linder.

**Project administration:** Johan Lundin, Harrison Kaingu, Nina Linder.

**Resources:** Johan Lundin, Harrison Kaingu, Nina Linder.

**Software:** Johan Lundin, Antti Suutala, Mikael Lundin.

**Supervision:** Johan Lundin, Harrison Kaingu, Nina Linder.

**Validation:** Johan Lundin, Antti Suutala, Oscar Holmström, Samuel Henriksson, Severi Valkamo, Felix Kinyua, Nina Linder.

**Visualization:** Johan Lundin, Antti Suutala, Nina Linder.

**Writing – original draft:** Johan Lundin, Antti Suutala, Oscar Holmström, Samuel Henriksson, Severi Valkamo, Nina Linder.

**Writing – review & editing:** Johan Lundin, Antti Suutala, Oscar Holmström, Samuel Henriksson, Severi Valkamo, Harrison Kaingu, Felix Kinyua, Martin Muinde, Mikael Lundin, Vinod Diwan, Andreas Mårtensson, Nina Linder.

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
