## [Decision Letter · Decision Letter 0]

9 Oct 2023

Dear Professor Lundin,

Thank you very much for submitting your manuscript "Diagnostics of soil-transmitted helminths with digital mobile microscopy and artificial intelligence in a resource-limited setting" for consideration at PLOS Neglected Tropical Diseases. As with all papers reviewed by the journal, your manuscript was reviewed by members of the editorial board and by several independent reviewers. In light of the reviews (below this email), we would like to invite the resubmission of a significantly-revised version that takes into account the reviewers' comments. 

We cannot make any decision about publication until we have seen the revised manuscript and your response to the reviewers' comments. Your revised manuscript is also likely to be sent to reviewers for further evaluation.

Sincerely,

Luc E. Coffeng, MD PhD

Guest Editor

Eva Clark

Section Editor

Reviewer's Responses to Questions

**Key Review Criteria Required for Acceptance?**

**Methods**

-Are the objectives of the study clearly articulated with a clear testable hypothesis stated?

-Is the study design appropriate to address the stated objectives?

-Is the population clearly described and appropriate for the hypothesis being tested?

-Is the sample size sufficient to ensure adequate power to address the hypothesis being tested?

-Were correct statistical analysis used to support conclusions?

-Are there concerns about ethical or regulatory requirements being met?

Reviewer #1: The scientific obejctive of this work and the novelty is not clear enough. This should be improved upon accordingly. 

The study design is good enought.

The population is clearly described and appropriate for the hypothesis being tested.

Statistical analyisis used are satisfactory

Appropriate ethical regulatory requirement has been met.

Reviewer #2: Methods

Line 121 and 122; do we need the abbreviations TUM and NACOSTI – they are only used once. 

Line 128: no need to capitalize albendazole

Line 134: please refer to STARD-AI (see general comment) and adapt the manuscript were required.

Line 139: there were any inc/exclusion criteria

Fig 1: see STANDARD AI

Line 157: why can people other the reviewers not access this. I would like to encourage the authors to add them.

Line 165: there is systematic bias in the study design (see general comment)

Line 177: this exclusion of samples also needs to be discussed; in a way you are favoring for clear slides

Line 193: how long does it take to scan one slide

Line 212: per slide / all slides

Fig 2: Can you confirm the scanning automated?

Lines 223 – 232: this might require a review by a AI-expert

Lines 245: not clear whether all digitized slides (line 245) or only discrepancies (line 186-189)

Fig 4: see remark on definition, of ground truth

Line 277: can one indicate the number of slides per STH specfies; per set (training, valiation).

Line 294: please define the level of confidence

Statistical analysis

Line 309: The word ‘comparable’ suggests an equivalence testing

Line 314: not sure where the correlation refers to; I am assuming EPG; see also general comment.

Reviewer #3: The methods used in this paper are appropriate.

**Results**

-Does the analysis presented match the analysis plan?

-Are the results clearly and completely presented?

-Are the figures (Tables, Images) of sufficient quality for clarity?

Reviewer #1: Details contained in the attachement

Reviewer #2: Results

See general comments on how results are represented

Would welcome more alignment with the hypothesis (‘comparable to the ground thruth). 

Table 3. Replace AL by Ascaris lumbricoides

Reviewer #3: Results are clearly presented, and they correspond to the aims of this study.

**Conclusions**

-Are the conclusions supported by the data presented?

-Are the limitations of analysis clearly described?

-Do the authors discuss how these data can be helpful to advance our understanding of the topic under study?

-Is public health relevance addressed?

Reviewer #1: Details contained in the attachement

Reviewer #2: Discussion 

Line 400 – compared to the ground truth (manual microscopy) I would argue that the accuracy is not high; of course reviewing the discrepancies it does. Again, it might help to re-define ground truth. 

Line 448 – perhaps it would be good to emphasize a health centre looks like; is indeed representative for a setting with poor infrastructure

How do the authors feel about the extrapolation of the findings to other setting

Reviewer #3: Yes

**Editorial and Data Presentation Modifications?**

Reviewer #1: Details contained in the attachement

Reviewer #2: See general comments

Reviewer #3: (No Response)

**Summary and General Comments**

Reviewer #1: (No Response)

Reviewer #2: Lundin and colleagues assessed the performance of digital mobile microscopy supported with AI to detect and quantify STH eggs in KK thick-smear. For this, they conducted a study in a health center (Kenya), where stool samples for school-aged children (SAC) were processed by both manual microscopy and AI. Based on the study results, they conclude that digital mobile microscopy supported with AI holds promise as a diagnostic standard in health care settings. 

General comments

1/ It is not clear to me where the authors position their invention. Is it to support program-decision making (determine the frequency and intensity of MDA), as highlighted in the introduction, or to support diagnosis in the health center, as concluded in the abstract. Both are not only quite distinct in decision making (population level vs. individual treatment) and field use (point-of-sampling vs. point of care), I also doubt whether Kato-Katz thick smear is the diagnostic standard in a health center. Too often this is just a direct smear, to ensure detection of a wider spectrum of parasites, including protozoa. 

2/ The study design has some an important flaw: the slides were first subjected to manual microscopy, after which they were scanned. Given a time bounded procedure such as Kato-Katz (hookworm eggs disappear), this has some consequences. This might indeed explain why some hookworms were missed by DSL. Yet, it might also explain why some Ascaris and Trichuris eggs were detected by DSL but not by manual microscopy. While hookworm eggs disappear over time due to the stool clearing process, Ascaris and Trichuris eggs might become more visible over time. In other words, the improved accuracy might be solely due to a difference in clearing time. This is in particular when slides were immediately examined (line 165), and hence there is slides were not cleared at all. Study cannot be redone, but this should be explicitly mentioned in the discussion. 

3/ I feel that result section misses some clarity/info. First, the word ground truth in an AI setting for manual reading is confusing. My interpretation of ground truth is that whole digitized slides are manually and independently annotated by preferable multiple independent experts. Ground truth is now used for manual microscopy (Table 3). Second, I would also propose to add a Table that represents that results after visual review of the digitized samples. This will help the reader to reasoning of the conclusions about the sensitivity/specificity. A sensitivity of 70 – 92% compared to manual microscopy is not really an improvement, whereas the visual review indicates that manual microscopy missed most of the positive samples detected by DLS. Third, I would welcome a scatter plot plotting the EPG for manual microscopy as a function of those obtained by DLS that could replace Table 4. Such a graph would give more insights in the data: (i) correlation in EPG between both diagnostics; (ii) where they fail compared to the other one (dots on the y-/x- axis), (iii) agreement in classification of infection intensity, and (iv) highlight the visual review results (this might require some thought). 

4/ Appreciate the attention the authors give to report according to the STARD-guidelines. Given the involvement of AI, I would encourage them to also consult the STARD-AI guidelines (Sounderajah et al., 2021; BMC OPEN Protocol), and to adapt the manuscript accordingly. 

5/ Author summary: propose to use more lay language. For example, intestinal worms instead of STHs.

Reviewer #3: Overall, the manuscript adds value to the literature for the detection of eggs of STH in developing countries by providing confidence of removing skilled personnel for microscopy. The methods are appropriate, results are presented nicely and interpreted carefully. Therefore, I recommend this paper for publication after a minor revision due to the following reasons:

Abbreviations used in some tables need to explained in footnotes.

Scale bars should be given for microscopic images

References should be checked for style and formatting as per the journal's guidelines.

PLOS authors have the option to publish the peer review history of their article (what does this mean?). If published, this will include your full peer review and any attached files.

Reviewer #1: No

Reviewer #2: No

Reviewer #3: No
---

## [Decision Letter · Decision Letter 1]

18 Jan 2024

Dear Professor Lundin,

Thank you very much for submitting your manuscript "Diagnostics of soil-transmitted helminths with digital mobile microscopy and artificial intelligence in a resource-limited setting" for consideration at PLOS Neglected Tropical Diseases. As with all papers reviewed by the journal, your manuscript was reviewed by members of the editorial board and by several independent reviewers. The reviewers appreciated the attention to an important topic. Based on the reviews, we are likely to accept this manuscript for publication, providing that you modify the manuscript according to the review recommendations. 

Dear authors,

Thank you for addressing the three reviewer's comments. Two of the reviewers still have some points for (minor) revision, which I would like to ask you to address. Please note that one of the reviewers has listed their points in an attached pdf file (also downloadable from the online editorial system).

Sincerely,

Luc E. Coffeng, MD PhD

Academic Editor

Eva Clark

Section Editor

Dear authors,

Thank you for addressing the three reviewer's comments. Two of the reviewers still have some points for (minor) revision, which I would like to ask you to address.

Reviewer's Responses to Questions

**Key Review Criteria Required for Acceptance?**

**Methods**

-Are the objectives of the study clearly articulated with a clear testable hypothesis stated?

-Is the study design appropriate to address the stated objectives?

-Is the population clearly described and appropriate for the hypothesis being tested?

-Is the sample size sufficient to ensure adequate power to address the hypothesis being tested?

-Were correct statistical analysis used to support conclusions?

-Are there concerns about ethical or regulatory requirements being met?

Reviewer #1: The objectives of the study are clearly articulated. Further comments are included in the attachments.

Reviewer #2: (No Response)

**Results**

-Does the analysis presented match the analysis plan?

-Are the results clearly and completely presented?

-Are the figures (Tables, Images) of sufficient quality for clarity?

Reviewer #1: (No Response)

Reviewer #2: (No Response)

**Conclusions**

-Are the conclusions supported by the data presented?

-Are the limitations of analysis clearly described?

-Do the authors discuss how these data can be helpful to advance our understanding of the topic under study?

-Is public health relevance addressed?

Reviewer #1: (No Response)

Reviewer #2: (No Response)

**Editorial and Data Presentation Modifications?**

Reviewer #1: (No Response)

Reviewer #2: (No Response)

**Summary and General Comments**

Reviewer #1: (No Response)

Reviewer #2: Lundin and colleagues have thoroughly revised the manuscript and addressed all my major comments. I only have a few minor comments suggestions that the authors may want to consider to further improve the manuscript. 

Abstract

Line 21: there is a conflict in words; NTDs refers to diseases, while soil-transmitted helminths refer to the pathogens; better to use ‘soil-transmitted helminthiasis’ instead

Line 24: WHO target is not to eliminate STH, rather the elimination of STH as a public health problem.

Line 47: I think there is broad consensus that this not perfect. So I am not sure whether you can make this conclusion. 

Author summary

Line 53: stool instead of feces

Introduction:

See previous comment on diseases vs. pathogen

Line 73: they are all roundworms, Ascaris is sometimes referred as giant roundworm

Line 83: prefer to state ‘recommended method’ or ‘current diagnostic standard’ instead of ‘gold standard’

Line 96: FECPAKG2 instead of FECPAK (this the first generation)

Line 97: not all of these have a higher sensitivity.

Line 118: ‘Kato-Katz thick smear’ instead of ‘faecal samples’

Line 119/line218: ‘smears’ instead of ‘samples’; check throughout the manuscript

Methods

Lin 161: 

Line 171: mL instead of ml

Line 274: n = 388?

Line 264: Fig 3 instead of Figure

Line 343: what does epsilon means in these formulae?

Table 2: omit the row for deep learning; this is made clear in the caption

Line 405: Table S1 instead of S Table 1

Line 407: Fig S1 instead of S1 Figure

Lines 437 – 441: the core question is whether the classifications of infection intensity based on DLS would be different when using manual microscopy. This is more relevant than the difference in mean in EPG. I would like to see this in the body of the text instead of putting into it in supplementary info 

Discussion 

Line 453: avoid the use of gold standard

Line 492: the way forward would be either to randomize the KK slide over manual or WSI; or to prepare a duplicate slide which are then processed in parallel. 

Line 518: put genus names in italic; note that the performance in the TPP are absolute, and not relative to the manual reading of KK, and hence may want to tune down the conclusions a bit down.

PLOS authors have the option to publish the peer review history of their article (what does this mean?). If published, this will include your full peer review and any attached files.

Reviewer #1: No

Reviewer #2: No

Figure Files:

Data Requirements:

Reproducibility:

References

---

## [Decision Letter · Decision Letter 2]

28 Feb 2024

Dear Professor Lundin,

We are pleased to inform you that your manuscript 'Diagnostics of soil-transmitted helminths with digital mobile microscopy and artificial intelligence in a resource-limited setting' has been provisionally accepted for publication in PLOS Neglected Tropical Diseases.

Best regards,

Eva Clark, M.D., Ph.D.

Section Editor

Eva Clark

Section Editor

Reviewer's Responses to Questions

**Key Review Criteria Required for Acceptance?**

**Methods**

-Are the objectives of the study clearly articulated with a clear testable hypothesis stated?

-Is the study design appropriate to address the stated objectives?

-Is the population clearly described and appropriate for the hypothesis being tested?

-Is the sample size sufficient to ensure adequate power to address the hypothesis being tested?

-Were correct statistical analysis used to support conclusions?

-Are there concerns about ethical or regulatory requirements being met?

Reviewer #2: yes

Reviewer #3: (No Response)

**Results**

-Does the analysis presented match the analysis plan?

-Are the results clearly and completely presented?

-Are the figures (Tables, Images) of sufficient quality for clarity?

Reviewer #2: yes

Reviewer #3: (No Response)

**Conclusions**

-Are the conclusions supported by the data presented?

-Are the limitations of analysis clearly described?

-Do the authors discuss how these data can be helpful to advance our understanding of the topic under study?

-Is public health relevance addressed?

Reviewer #2: yes

Reviewer #3: (No Response)

**Editorial and Data Presentation Modifications?**

Reviewer #2: (No Response)

Reviewer #3: (No Response)

**Summary and General Comments**

Reviewer #2: All comments are adessed.

Reviewer #3: (No Response)

PLOS authors have the option to publish the peer review history of their article (what does this mean?). If published, this will include your full peer review and any attached files.

Reviewer #2: No

Reviewer #3: No

---

## [Editor Report · Acceptance letter]

21 Mar 2024

Dear Professor Lundin,

We are delighted to inform you that your manuscript, "Diagnosis of soil-transmitted helminth infections with digital mobile microscopy and artificial intelligence in a resource-limited setting  ," has been formally accepted for publication in PLOS Neglected Tropical Diseases.

Best regards,

Shaden Kamhawi

co-Editor-in-Chief

Paul Brindley

co-Editor-in-Chief
